# Under Stress: Searching for Genes Involved in the Response of *Abies pinsapo* Boiss to Climate Change

**DOI:** 10.3390/ijms25094820

**Published:** 2024-04-28

**Authors:** Irene Blanca-Reyes, Víctor Lechuga, María Teresa Llebrés, José A. Carreira, Concepción Ávila, Francisco M. Cánovas, Vanessa Castro-Rodríguez

**Affiliations:** 1Grupo de Biología Molecular y Biotecnología, Departamento de Biología Molecular y Bioquímica en Instituto Andaluz de Biotecnología, Universidad de Málaga, Campus Universitario de Teatinos, 29071 Malaga, Spain; ireneblanca@uma.es (I.B.-R.); m.llebres@uma.es (M.T.L.); cavila@uma.es (C.Á.); 2Department of Ecology, Universidad de Jaen, Campus Las Lagunillas s/n., 23009 Jaén, Spain; vlechuga@ujaen.es (V.L.); jafuente@ujaen.es (J.A.C.)

**Keywords:** conifers, Mediterranean forests, Spanish fir, forest decline, gene families, climatic change, dry season, wet season

## Abstract

Currently, Mediterranean forests are experiencing the deleterious effects of global warming, which mainly include increased temperatures and decreased precipitation in the region. Relict *Abies pinsapo* fir forests, endemic in the southern Iberian Peninsula, are especially sensitive to these recent environmental disturbances, and identifying the genes involved in the response of this endangered tree species to climate-driven stresses is of paramount importance for mitigating their effects. Genomic resources for *A. pinsapo* allow for the analysis of candidate genes reacting to warming and aridity in their natural habitats. Several members of the complex gene families encoding late embryogenesis abundant proteins (LEAs) and heat shock proteins (HSPs) have been found to exhibit differential expression patterns between wet and dry seasons when samples from distinct geographical locations and dissimilar exposures to the effects of climate change were analyzed. The observed changes were more perceptible in the roots of trees, particularly in declining forests distributed at lower altitudes in the more vulnerable mountains. These findings align with previous studies and lay the groundwork for further research on the molecular level. Molecular and genomic approaches offer valuable insights for mitigating climate stress and safeguarding this endangered conifer.

## 1. Introduction

Forest ecosystems play an essential role in global carbon fixation and therefore in the mitigation of climate change as well as in the preservation of biodiversity. Additionally, tree species are of outmost economic importance for the production of wood, as well as a source of many other products, such as secondary metabolites and a great variety of resins.

The impact of anthropogenic activities on global warming is currently one of the major concerns of governments and society. Overall, the main threats of climate change include increased temperatures and limited water resources. Forest ecosystems are extremely sensitive to these environmental disturbances, and climate-driven increases in temperature and drought have been recently identified as major causes of forest die-off [1,2]. The adverse effects of these environmental factors are particularly relevant in the Iberian Peninsula, one of the most vulnerable areas in Europe, where the consequences of climate change and the advance of desertification are currently perceptible. Mediterranean coniferous forests are already affected by climate change [3,4], and recent simulations indicate that the impact will be even greater in the near future, especially for Mediterranean Abies forests [5,6,7]. Foreseeable changes can lead to chronic stress imbalances due to high temperatures and severe drought, which can cause forest decline, favoring greater sensitivity to pathogen attacks [8,9]. The conservation of forest stands is threatened, and adjustments in forest management will be necessary to counteract these adverse effects. In this context, it would be advantageous to have indicators of the adaptive capacity of trees and their tolerance to biotic and abiotic stresses [10,11]. However, the potential of forest trees to adapt to environmental stresses induced by climate change is largely unknown. Moreover, due to the experimental difficulties inherent in research on forest species, advances in molecular studies of the response to environmental stresses have not been as rapid as advances in models of herbaceous plants [12]. Therefore, it can be concluded that the global warming predicted for the coming years will strongly affect the performance of forests and will require changes to be made in forest management and conservation practices that can be assisted by genomics and biotechnology [13].

Pinsapo fir, also known as Spanish fir (*Abies pinsapo* Boiss.), is a tree in the family Pinaceae whose distribution is restricted to the southern mountains of the Iberian Peninsula. It is a relict species from the last ice age, and its habitat was greatly narrowed after the retreat of the ice. Spanish firs are usually found at altitudes of approximately 1000 to 1800 m in areas with a climate that includes rainy weather in winter and spring and long dry periods in summer. Two related fir species are also distributed throughout northern Morocco: *Abies maroccana* and *Abies tazaotana*.

Although pinsapo fir has limited economic importance due to the difficulties associated with its repopulation and the low quality of its wood, its importance is fundamentally ecological. It is a unique species, a remnant of the southern fir trees that disappeared when the last glaciers of the Cenozoic era in the Mediterranean retreated, but that has persisted in this region owing to the particular climatic conditions.

Abiotic factors associated with global warming have affected pinsapo forests to a great extent, directly or through synergistic interaction with land-use change [14,15,16]. Moreover, current forecasts indicate that, among the whole group of circum-Mediterranean fir species, the present situation could become even worse in the future for Spanish forests [6]. The molecular mechanisms underpinning the survival and adaptation of conifers, such as Spanish fir, are largely unknown, mainly because of the limited genomic resources that are currently available for these types of studies. However, in the last two decades, considerable progress has been made in understanding the molecular and cellular processes that control the growth and survival of trees through the use of new biological technologies [13,17]. Advances in molecular technologies such as massive DNA sequencing (next-generation sequencing, NGS) are having an increasing impact on research programs aimed at solving practical problems in the forestry sector [18,19,20,21,22,23,24].

A reference transcriptome for *A. pinsapo* was recently assembled that contains a large collection of full-length transcripts [25]. This genomic resource enables functional genomic studies to facilitate the identification and subsequent molecular analysis of relevant proteins potentially involved in the response of trees to increased temperatures and drought. In the present study, the structure of major gene families associated with the response to increased temperature and water stress was explored. Candidate genes were identified and selected for further expression analyses in growing trees in the forest under natural conditions. The results presented here revealed the differential seasonal expression patterns of these genes across various tissues in *A. pinsapo* from distinct geographical areas exposed to dissimilar weather conditions. These findings hold the potential to enhance our understanding of how *A. pinsapo* adapts to environmental stressors under natural conditions. Moreover, the acquired knowledge is relevant for further examining how environmental changes influence transcriptome dynamics. These molecular and genomic approaches will be of supreme importance for assisting the development of adaptive forest management strategies in the vulnerable forests of the southern Iberian Peninsula to fight against the adverse impacts of climate change.

## 2. Results

### 2.1. The Complex Gene Families of LEA and HSPs in A. pinsapo

In the present study, structural and expressional analyses of several gene families associated with plant responses to water scarcity and increasing temperature, two of the major factors influenced by current climate change, were performed. A reference transcriptome, including a large collection of unique full-length transcripts, was recently documented for *A. pinsapo* [25]. This genomic resource has been used to investigate the structure of the gene family encoding LEA and dehydrin proteins, and at least 65 unique transcripts belonging to this family were identified (Appendix A). Figure 1 shows a phylogenetic analysis of LEA sequences from pinsapo and other conifers, revealing their clustering in eight different clades (A–H) according to the nomenclature of [26], which is based on the occurrence of specific protein motifs in their primary structure (Appendix A). To facilitate the identification of the *A. pinsapo* sequences derived from the present work and those retrieved from Norway spruce, the former sequences are marked with green diamonds in the figure and the latter sequences with blue diamonds. Notably, most LEA type II proteins, also known as dehydrins, were grouped in the H clade, which is characterized by the presence of domain 1, corresponding to the well-conserved segment K (Appendix A).

Another main factor associated with climate change is the progressive increase in temperature. Some structural genes involved in coping with this critical environmental factor are those encoding heat shock proteins (HSPs), a large group of molecular chaperones assisting in the folding and/or refolding of polypeptides denatured by heat and other stressful factors. The inventory of such proteins in *A. pinsapo* has also been determined. Appendix A shows that the multigene family of pinsapo small heat shock proteins (sHSPs) comprises at least 36 members, most of which are of potential cytosolic localization, as inferred from the phylogenetic analysis presented in Figure 2. Additionally, possible mitochondrial, plastidic, and peroxisomal sHSPs, along with several proteins associated with the endoplasmic reticulum, were also identified. Similarly, for HSP70, a comparable number of sequences were identified (36, Appendix A), while the other three families of HSPs consisted of a smaller number of members: HSP60 (30, Appendix A), HSP90 (17, Appendix A), and HSP100 (13, Appendix A).

### 2.2. Selection of Candidate Genes and Trees Growing under Natural Conditions

The next step in this work involved the selection of candidate genes to study how their expression might change in response to drought and increased temperature. To achieve this goal, a prior study focused on Norway spruce (*Picea abies*) was used as a reference [27]. The choice of this previous study had the added advantage of providing a readily available extensive catalog of LEA and HSPs (see Figure 1 and Figure 2) since Norway spruce was the first conifer whose genome sequence was made available [28]. Several genes from Norway spruce exhibiting tissue-specific expression (roots and needles) in response to mild and severe drought, as well as in response to reirrigation, were selected. Figure 3 shows a combined heatmap of the expression levels of these genes under different conditions, along with the clustering of their corresponding orthologs identified in *A. pinsapo*. Additionally, Figure 3 includes a list of the accession codes for P. abies and *A. pinsapo* (Appendix A). Given that the full-length sequences of all the selected transcripts were available in the pinsapo transcriptome, specific oligonucleotides were designed for cDNA amplification (Appendix A) and for the quantitative determination of transcript levels in the trees.

The major aim of this study was to determine the expression levels of these selected candidate genes in adult trees growing in forests under natural conditions. Figure 4 shows the location of the “Sierra de las Nieves” National Park in southern Spain on the left, where the largest population of Spanish fir in the world grows. Two different sites that were in the geographical vicinity but exhibited contrasting climate characteristics were selected for sampling: Puerto Saucillo in Yunquera (YUN) and Cañada del Cuerno in Ronda (RON) (left panel). To examine the impact of climatic variations on the expression of the selected candidate genes, the samples were harvested from adult specimens (30-year-old trees) that were located in the forests at different altitudes RON (1734 m) and YUN (1196 m), where samples were collected during two seasonal periods: wet, in late spring and dry in early autumn (left lower panel). The organs sampled at these two locations were needles at the top of the trees (TN), needles at the base of the trees (BN), and root tissues (R). Seasonal variations in various climatic parameters clearly define well-differentiated dry and warm seasons in the region with maximal temperatures and minimum water content in the soil as a consequence of the lack of rainfall in summer. This dry period alternates with a rainy and cold period during winter (Figure 4, right panels). Interestingly, and despite their proximity, significant differences were observed in the seasonal variations in temperature between the YUN and RON forests. The same held true for soil water content, which was always greater in RON than in YUN, except the values observed during July, August and September, which reached quite similar minimal levels at both locations (Figure 4, right panel).

### 2.3. Changes in the Expression Levels of the ApDHN and ApLEA Genes in Response to Increasing Temperature and Water Shortage under Natural Conditions

The relative expression levels of the selected genes potentially responding to changes in water stress and temperature were quantitatively determined in the previously described Spanish fir samples, as described earlier. According to the heatmap shown in Figure 3, three DHN genes—ApDHN6, ApDHN11, and ApDHN17—were selected; all of these genes were included in the H clade of the phylogenetic tree (Figure 1). In the samples harvested in September (end of the dry season, before the first autumn rains), the ApDHN11 gene was expressed at low levels in tree-crown top and basal needles of trees at both localizations, YUN and RON (Figure 5). However, pronounced differences were observed in the transcript levels detected in the roots, with a much greater abundance in the forest located at lower altitudes during this dry seasonal period. In May (spring, which is the peak of vegetative growth within the wet season), the highest expression level of the gene was also detected in the roots of YUN, but the relative levels of the transcripts were much lower than those observed in the dry season and nearest to the levels detected in green tissues (Figure 5, ApDHN11). The expression patterns of two additional genes encoding dehydrins, ApDHN6 and ApDHN17, were also examined. In early autumn, increased transcript abundance was observed in the roots of trees sampled from both the YUN and RON locations in comparison with the values determined in the top (TNs) and basal (BNs) needles. A similar expression pattern was also observed in spring, although the relative expression levels observed in green tissues were considerably lower than those observed in the dry season (Figure 5, a comparison of autumn versus spring). In contrast, transcripts encoding ApDHN17 were highly abundant in top needles (TNs) by the end of the dry season at high altitude (RON), while in the wet season, this gene was expressed mainly in the roots of trees from the drier low altitude (YUN).

The expression patterns of four selected members of the LEA multigene family were also investigated: ApLEA8, ApLEA15, ApLEA28, and ApLEA30 (Figure 5). These four genes are classified into different clades: ApLEA8, in clade B; ApLEA15, in clade C; and ApLEA28 and ApLEA30, which are both in clade D (Figure 1), suggesting that they are phylogenetically unrelated. In the dry season (early autumn), limited expression levels of ApLEA8 were detected in the photosynthetic and non-photosynthetic tissues of trees at low and high altitude, while increased transcript abundance was apparent in spring, particularly in the needles of trees growing in the RON forest. ApLEA15 was exclusively expressed in the needles in both seasons (dry and wet), with higher levels of transcripts detected in tree-crown top needles. Finally, substantial transcript levels of the ApLEA28 and ApLEA30 genes were exclusively detected in needles collected from YUN in the dry season, and the transcripts were barely detected in the RON forest. Interestingly, the expression levels were almost undetectable in all tissues from both origins in samples corresponding to the wet season.

### 2.4. Changes in the Expression Levels of HSP Genes in Response to Increasing Temperature and Water Shortage under Natural Conditions

HSPs consist of different families of molecular chaperones involved in buffering the denaturing effects caused by a broad array of environmental stresses, including heat. Increased expression levels of the selected ApHSP60-24 gene were detected in the young needles and roots of trees located both in YUN and RON in the dry season (autumn), whereas no significant changes in expression were perceptible among samples harvested in spring (Figure 6). In contrast, the two selected genes of the sHSP subfamily, ApsHSP8 and ApsHSP12, exhibited similar patterns of gene expression, with peak values of transcript abundance detected in the roots of trees sampled in YUN. Interestingly, no major differences in gene expression were found between the dry and wet seasons at this specific location. Much lower levels of ApsHSP8 and ApsHSP12 transcripts were detected in the different tissues of trees located in the RON forest. Two candidate genes from the subfamily HSP70, ApHSP70-1 and ApHSP70-34, exhibited contrasting gene expression patterns in the analyzed conditions and the tissues that were analyzed (Figure 6). ApHSP70-1 was expressed mainly in the roots of trees from YUN and RON, independent of altitude (YUN or RON) and the season (dry or wet) of sampling. In contrast, ApHSP70-34 showed predominant expression in the green needles of the RON source in autumn; however, increased levels of transcript abundance were recorded in the wet season in the YUN forest. The gene encoding this chaperone was expressed in roots only in spring. Increased levels of ApHSP90-9 expression were observed in the tree-crown top needles and in the roots of specimens sampled in both pinsapo forests, independent of the season (dry or wet). Conversely, for the candidate member of the HSP100 subfamily, increased levels of ApHSP100-2 expression were restricted to the roots of trees growing in YUN (Figure 6). ApHSP 60-23 and ApHSP100-3 members have shown buffering expression profiles, although HSP70-9 exhibits expression levels 100 times higher in all sampled tissues and in both locations and seasons of harvest (Appendix A).

## 3. Discussion

Currently, major factors associated with climate change include increased ambient temperatures and rainfall shortages, which exacerbate severe drought in many areas. These environmental changes are challenging the adaptation of coniferous species to their habitats and forcing them to cope with unexpected biotic and abiotic stresses. Simulated predictions for the next several decades reveal scenarios of particular vulnerability in endangered conifer species, such as Mediterranean firs, particularly for *A. pinsapo* endemics from the southern Iberian Peninsula, both if forecasts are based on niche distribution models [30] or in process-based tree-growth trends [6]. The multigene families encoding LEA and HSP proteins are well documented in the scientific literature as integral parts of the response of plants to abiotic stress. The members of these gene families in *A. pinsapo* were selected to accomplish this study in the forest populations of the “Sierra de las Nieves” National Park.

LEA proteins were first described in angiosperms as possessing an essential function in the desiccation of seeds during the last steps of embryogenesis [31]. Members of the large and diverse gene family of LEA are involved in the protection of plant cells against the harmful effects of a wide range of abiotic stresses, including drought [32]. The structure and potential function of several members of the LEA family have been previously described in various conifer species, including *Pinus pinaster* [33,34], *Picea glauca* [35], and *Pinus tabuliformis* [36,37]. Genomic evidence indicates an expansion of this gene family in the coniferous family Pinaceae, including the LEA group II, also known as dehydrins, suggesting that the diversification of the family has the potential function of enhancing drought tolerance in conifers [38]. These prior findings further support the suitability of members of the LEA family as potential markers for studying the effects of increasing aridity on pinsapo forests.

In recent decades, the effects of climate change, marked by recurrent periods of drought and prolonged warming, have threatened the growth and viability of pinsapo forests in their mountain climatic refuge in southern Spain, but especially in Sierra de las Nieves [7,15]. This situation has been particularly aggravated in recent years, during which unprecedented increases in temperature and persistent drought periods have taken place in southern Spain. Limited water availability and high temperatures, in interaction with increasing stand-level intra-specific competition derived from strict protection measures, have been determined to be the main abiotic factors involved in the dieback and decline of pinsapo forests [16]. Furthermore, Spanish fir populations located at lower altitudes in their distribution area are more vulnerable to these climate-driven impacts than those located at higher altitudes [8,39]. All these previous studies involved the sampling of adult trees that were growing in the forest in lower and upper ecotones and subjected to contrasting microclimatic conditions despite being located nearby (Figure 4).

It has been shown that local conifer populations are well-adapted to microenvironmental conditions [40]; therefore, their adaptation to climatic changes will depend largely on the genetic variability existing in such populations. Here, it is hypothesized that approaches based on molecular biology and genomics can provide insights into the molecular basis of pinsapo forests’ response to abiotic stresses, as well as a better understanding of their adaptive capacity. Therefore, the identification of markers for forest resilience against these adverse factors is of the utmost importance [41]. The identification of genes that respond differently to high temperature and drought, as achieved in the present work, represents a valuable step in this direction.

The study of the climate-driven effects on *A. pinsapo* gene expression has been previously undertaken using experimental approaches and treatments under controlled environmental conditions [42]. However, in this work, we have assessed the issue by field-sampling trees growing in their natural habitats. On the other hand, most of the existing knowledge on conifer species transcriptomic responses to temperature and water availability is often based on micro- and mesocosm studies using seedlings or ramets instead of adult trees [43,44], including those conducted with pinsapo saplings [42]. The results derived from such studies are, unquestionably, of high value for obtaining insights into the molecular basis of tree responses to various stresses, especially under extreme conditions. However, the derived data do not necessarily reflect the actual impacts that increasing temperatures and drought have in nature, especially on adult trees. Thus, to analyze the in situ gene expression responses to the current environmental stresses in the pinsapo forests, in the present work, different tissues from 30-year-old trees growing at 2 × 2 contrasting elevations and seasons were examined (Figure 4).

Overall, the expression patterns of the selected candidate genes in *A. pinsapo* differed significantly from those observed in response to mild and severe drought in mesocosm in *P. abies* (Figure 3, Figure 5 and Figure 6). These findings indicate a substantial difference in the gene expression patterns of phylogenetically related genes between experimentally controlled and natural conditions, underscoring the challenges of extrapolating results between these two situations [27] (and present work). Even considering that two different conifer species were compared, it is noteworthy to mention the relative expression levels of several transcripts, such as ApDHN6, ApDHN11, ApsHSP8, and AsHSP12, which were found to be more abundant in the roots than in the needles of more vulnerable trees (YUN). This suggests that these genes can be considered true orthologs of their corresponding counterparts in *P. abies*, and are potentially involved in the primary stress response, which is consistent with what was reported previously [27]. Another potential ortholog with enhanced expression levels in the crown-top needles of vulnerable trees was ApLEA15 (Figure 5). Nevertheless, the observed differences may also reflect the distinct age of the plants that were analyzed—saplings in the case of *P. abies* and adult trees in that of *A. pinsapo*.

The transcript abundances of several candidate genes, such as ApHSP60-23, ApHSP60-24, ApHSP70-9, and ApHSP100-3 (Figure 6 and Appendix A), did not significantly change among samples from different tissues or locations. These results suggest that these genes may serve housekeeping functions in Spanish firs. Members of the HSP60 family are highly conserved proteins with chaperone functions that help with polypeptide folding and subunit assembly mainly in mitochondria and chloroplasts [45]. The majority of genes belonging to this HSP60 family are expressed uniformly in plants, as is the case for ApHSP60-24 in *A. pinsapo*, which is expressed from the base to the top of the tree at similar levels (either in the roots or in needles from the tree-crown top or the base). Furthermore, there was no difference in its expression concerning either the season of the year or the forest location (high or low altitude).

Members of the HSP70 family mainly function as plant molecular chaperones and may also participate in stress signaling, potentially collaborating with HSP100 in the targeting and degradation of protein aggregates [46]. Therefore, the nearly constitutive expression of HSP70 and HSP100 is likely essential for the turnover (degradation or reactivation) of key enzymes involved in metabolic pathways required to maintain cellular homeostasis in response to stress.

When considering the expression patterns exclusively in *A. pinsapo,* genes exhibiting enhanced expression levels in the roots, such as ApDHN6, ApDHN11, ApsHSP8, ApsHSP12, ApHSP70-1, and ApHSP90-9, and other genes preferentially expressed in needles, such as ApLEA15, ApLEA28, and ApLEA30, were found. These findings suggest a distinctive tissue-specific response to environmental changes. In fact, roots are known to be the primary responders to abiotic stresses, initiating systemic signaling to the aboveground parts of trees [47]. Taken together, the above data suggest that the adaptation of pinsapo firs to climate change may involve mechanisms that act at different developmental levels and in different organs, which is consistent with the findings of previous studies on other conifer species [27,44]. For instance, the genes associated with the drought response in pinsapo roots encode dehydrins, which are Group II LEA proteins that play crucial roles in the response of plants to drought, cold and other environmental stresses [48]. Conversely, in needles, the transcripts differentially expressed in the dry season encoded proteins belonging to LEA groups other than dehydrins. Previous molecular studies have consistently reported that roots are more sensitive to the effects of environmental stresses than aboveground tissues; therefore, changes in gene expression are first detectable in the roots [27,49]. The enhanced expression of sHSP genes (ApsHSP8 and ApsHSP12) in the roots also suggests a greater thermotolerance capacity in the trees. In fact, increased thermotolerance and chaperone activity in poplar have been reported to be mediated by the overexpression of a major sHSP protein [50]. Moreover, a sHSP was recently identified as a key regulator of a gene network involved in protective roles to heat stress via the upregulation of a variety of molecular chaperones and transcription factors controlling responses to abiotic stress [51].

A representation of the major differences found in the gene expression patterns of stress-related genes is outlined in Figure 7. Trees at lower altitude (YUN) had more differentially expressed genes than trees located at higher altitude (RON), reflecting a significant impact of climate change on the most vulnerable population. This difference is particularly accentuated in both the roots and needles sampled in early autumn, before the first post-summer rains, at the moment when trees had been experiencing intense drought and high temperatures during the 3–4 months long dry season. These molecular data align with climatic conditions (Figure 4) and previous ecophysiological studies [9,52,53]. It should be interesting to analyze the expression patterns of stress-related genes in the next seasons to confirm the adaptive response of pinsapo firs to climate change. Furthermore, this study provides additional results that deserve discussion in this context. The enhanced expression levels of ApHSP70-1 and ApHSP90-9 in the roots of trees from both the YUN and RON forests suggest the possible involvement of these genes in the basal response to stress. Moreover, this response not only occurs in both populations but also seems to be independent of the season (wet or dry), and it is tempting to speculate that these two genes could be involved in signaling mechanisms that prevent further deleterious effects of environmental stresses and, therefore, contribute to the resilience of pinsapo forests against climate change [39,46]. In contrast to this, the expression of ApDHN6 appears to be specific to the dry season in both populations, and the expression of ApDHN11 specifically increased in YUN, the most vulnerable population at the driest and hotter sites. The expression pattern of ApDHN17 merits special attention because this gene was differentially expressed in the roots of YUN trees in the wet season, and interestingly, in the crown-top needles of RON trees in the dry season. These findings suggest the occurrence of a systemic signaling process from roots to the tree-crown photosynthetic tissues involved in the expression of this particular gene in both populations, which deserves further investigation in future work.

The identification of better-adapted populations of pinsapo fir is crucial for developing adaptation strategies to mitigate the effects of climate change, leveraging genomic resources. Studies on *Abies alba* populations have highlighted varying levels of adaptation to abiotic stress, suggesting the presence of complex evolutionary strategies [54,55]. However, further genomic studies are needed to determine how genetic variability in different populations supports phenotypic plasticity in the adaptation to climate-driven stresses. The data reported here suggest that exploring genetic variability in the structure and number of candidate genes at the genomic level in various populations is essential. This approach can aid in selecting genotypes that are better suited for reforestation programs in the most stressed areas, contributing to the recovery and conservation of *A. pinsapo.* The implementation of assisted migration programs, involving the transfer of individual trees from specific habitats to other habitats with better current or future microclimatic conditions, is one strategy. Another factor to be considered is the promotion of *A. pinsapo* stands with high canopy structural diversity and low intensity and asymmetric competition, which has been shown to enhance tree growth and water uptake in thinned stands [9], consistent with the proposed role of greater competition in forest decline and tree mortality [56,57]. Given the extended reproductive time required for these coniferous trees, the multiplication of the best-suited genotypes identified by using molecular markers and genomic data can be accelerated through vegetative multiplication via somatic embryogenesis (SE). Protocols for the efficient production of somatic embryos have been developed for several conifer species [58,59], and embryogenic cultures can be cryopreserved for long periods [60]. SE technology, in combination with molecular tools [61] (this work), could yield elite genotypes for the conservation of pinsapo forests. In addition, SE protocols are adaptable for automation and scaling up using bioreactors and robotic devices [62]. Given these considerations, the development of functional SE in endangered conifer species like *A. pinsapo* deserves urgent research efforts.

## 4. Materials and Methods

### 4.1. Plant Material

Plant material used for analysis was collected in the national park “Sierra de las Nieves”, which covers approximately 23,000 hectares in the province of Málaga (southern Spain) and houses the largest population of Spanish fir (about 2/3 of its complete distribution area). Two sites that were in the geographical vicinity within this area but exhibit contrasting climate characteristics were selected for sampling: Puerto Saucillo-Yunquera (lower ecotone, 1100–1200 m asl was situated at 36°43′ N, 4°58′ W; denoted as ”YUN” hereafter) and Cañada del Cuerno-Ronda (upper ecotone, 1700–1800 m asl was located at 36°41′ N, 5°01′ W; denoted as “RON” [53]. The monitoring of microclimatic conditions within each site, including air temperature (T, °C), relative air humidity (RH, %), and soil water content (SWC, % vol), was conducted using two dataloggers per trial. T-RH sensors per plot were positioned at a height of 1 m above the ground, while four SWC probes per plot were buried at a depth of 0.30 m. Temperature, soil water content, and rainfall in the two locations harvested and their interactions were analyzed via ANOVA using R (www.R-project.org (accessed on 25 April 2024)) (Appendix A). The instrumentation details can be found in [9]. Sampling occurred during two distinct periods, one in September and the second in May. During each field sampling session, 10 trees per site were randomly selected. Subsequently, needles located above 2–3 m above the ground (considered top needles), those at 1 m or less from the ground (considered base needles), and main roots were harvested. Samples were placed into a portable liquid nitrogen container and deposited in dry ice for transport to the laboratory, where they were stored at −80 °C until experiments were performed.

### 4.2. RNA Extraction

Total RNA extraction followed the procedure described in [63]. For total root RNA extractions, the manufacturer’s protocol from the NucleoSpin RNA Plant (Cat. No. 740120.50, Macherey-Nagel, Düren, Germany) was performed. The samples were initially quantified using a NanoPhotometer N60 (Implen) (Fisher Scientific, Leicestershire, UK) spectrophotometer, with purity assessed based on the 260/280 and 260/230 ratios. RNA integrity was checked via agarose gel electrophoresis and also determined in a Fragment Analyzer System using the Agilent DNF-472 HS RNA Kit (Agilent, CA, USA). Only samples with RIN >7 were selected for further analyses.

### 4.3. RT-qPCR Analysis

The cDNA synthesis utilized 0.5 μg of total RNA and the iScript™ cDNA Synthesis Kit (Bio-Rad, Hercules, CA, USA), following the manufacturer’s instructions. The qPCR primers were designed following the MIQE guidelines [64], and their details are provided in Appendix A. qPCRs were performed with 10 ng of cDNA, 0.4 mM of primers, and 2X SsoFast™ EvaGreen^®^ Supermix (Cat. No. 1725204, Bio-Rad, Hercules, CA, USA) in a total volume of 10 μL. Relative quantification of gene expression was conducted using the thermocycler CFX 384™ Real-Time System (Bio-Rad, Hercules, CA, USA). The qPCR program comprised an initial cycle of 2 min at 95 °C, followed by 40 cycles of 1 s at 95 °C and 5 s at 60 °C, with a melting curve ranging from 60 to 95 °C to confirm the specific amplification of each reaction. Analyses were performed as described by [20] using the MAK3 model in the R package qPCR [65]. Expression data were normalized with a reference gene ap_33 (ABC transporter F family member), which were previously validated for RT-qPCR experiments in [25]. Each qPCR analysis included three biological replicates and three technical replicates per sample. To determine statistical differences in the expression of candidates, Past 4.03 was used [29]. The data were initially analyzed using a Kruskall–Wallis test, and subsequently, upon identifying significant differences, a multiple comparisons test (Dunn posthoc) was performed.

### 4.4. Candidate Selection and Protein/Motif Identification for Drought Stress

Drought stress candidates LEA, DHN, and HSP proteins in *A. pinsapo* Boiss were identified by searching and aligning them with previously recognized genes associated with drought response in *P. abies* [27]. The orthologs of these selected candidates were then employed to conduct a comprehensive search, aiming to identify conserved protein domains and functions within differentially expressed transcripts in samples of pinsapo needles and roots (Appendix A).

To identify *A. pinsapo* LEA, DHN, and HSP proteins, sequence similarity searches were performed using the *P. abies* proteins ConGeniE database https://plantgenie.org/ (accessed on 22 April 2021) and BLASTP against our assembled *A. pinsapo* transcriptome [25]. The identification process also included the use of MEME 5.5.5 software (http://meme-suite.org/ (accessed on 25 April 2024)) to confirm the presence of conserved domains in their protein structures. The parameters used for discovering motifs were a distribution of any number of repetitions with a minimum of 6 and a maximum width of 50. To predict subcellular location, we use the DeepLoc tool [66].

### 4.5. Phylogenetic Analysis

Phylogenetic analyses were conducted on 65, 36, 30, 36, 17, and 13 protein sequences from ApLEA, ApsHSP, ApHSP60, ApHSP70, ApHSP90, and ApHSP100, respectively, in pinsapo (Appendix A). Additionally, sequences from *P. abies*, *Pinus taeda*, *A. pinsapo* were included for LEA phylogenetic analyses (Figure 1), while sequences from *Arabidopsis thaliana* were included for HSP sequences to examine subcellular location (Figure 2 for sHSP and Appendix A for HSP60, 70, 90 and 100). Sequences were aligned using default parameters in ClustalW [67] for LEA proteins and MUSCLE [68] for HSPs, and phylogenetic analysis employed the maximum parsimony method, utilizing a bootstrap approach with 1000 replications to assess the robustness of the inferred phylogeny. Amino acid substitutions were considered, and gaps or missing data were included across all sites. The analysis employed the subtree-pruning-regrafting (SPR) search method with 10 initial random trees, retaining up to 100 trees and utilizing 8 threads for computational efficiency. The trees were generated using MEGA X [69].

## 5. Conclusions

Little is known about the molecular response to climate-change-associated stresses in the Spanish fir (*A. pinsapo*), a conifer species of great ecological relevance that is in danger of extinction. In this work, a first approach was conducted to shed light on the molecular mechanisms governing the response to abiotic stress in trees within their natural ecosystem by studying gene families linked to increased temperature and water stress. Candidate genes within these families were identified, and their differential seasonal expression patterns were examined across various tissues of trees sampled from distinct geographical areas with dissimilar weather conditions. The data showed that trees growing at lower altitude possess a greater number of differentially expressed genes than trees located at elevated altitude, reflecting a major effect of climate change on the lower and most vulnerable populations. Furthermore, differences in gene expression were more pronounced in the roots and needles sampled in early autumn, when forests were affected by several months of drought and high temperatures during the long dry season. Further functional analysis of these genes will provide insights into the molecular mechanisms involved in the response of pinsapo fir to environmental stress. Taken together, these results hold the potential to significantly enhance our understanding of how local populations of *A. pinsapo* adapt to environmental stressors. Moreover, this knowledge will provide useful information for assisting adaptive forest management strategies in the forests of the southern Iberian Peninsula to mitigate the adverse impacts of climate change.

## Figures and Tables

**Figure 1 ijms-25-04820-f001:**
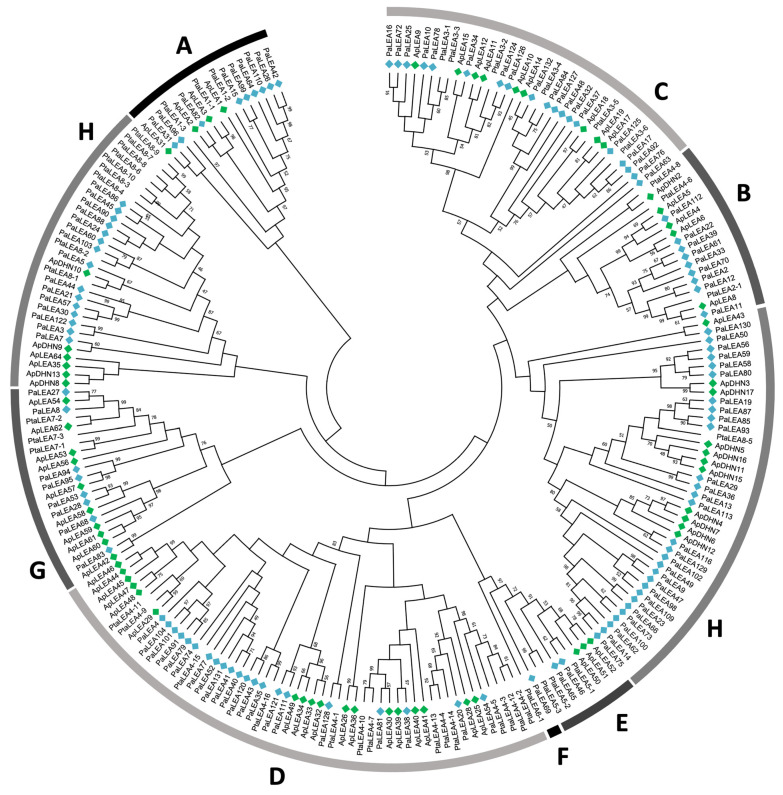
**Phylogenetic tree of plant LEA and dehydrin sequences**. The first two or three letters of each sequence correspond to the genera and species listed in Appendix A. The sequences from *A. pinsapo* and *P. abies* are marked by green and blue diamonds, respectively.

**Figure 2 ijms-25-04820-f002:**
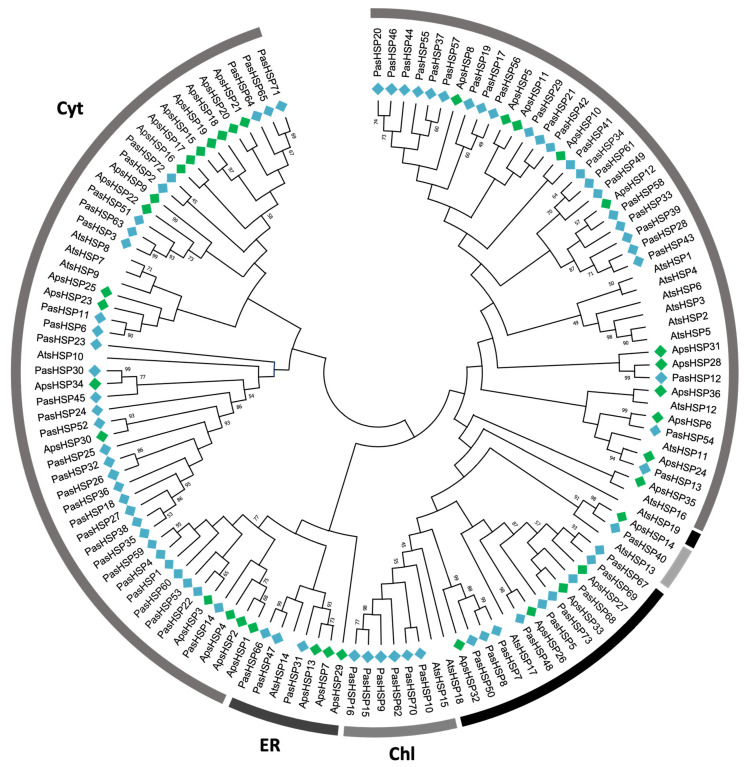
**Phylogenetic tree of plant sHSPs**. The first two or three letters of each sequence correspond to the genera and species listed in Appendix A. The sequences from *A. pinsapo* and *P. abies* are marked by green and blue diamonds, respectively. Subcellular locations of gene products are indicated with different gray colors. Cyt, Cytosol; ER, endoplasmic reticulum; Chl, chloroplast; Mit, mitochondria; and Per, peroxisome.

**Figure 3 ijms-25-04820-f003:**
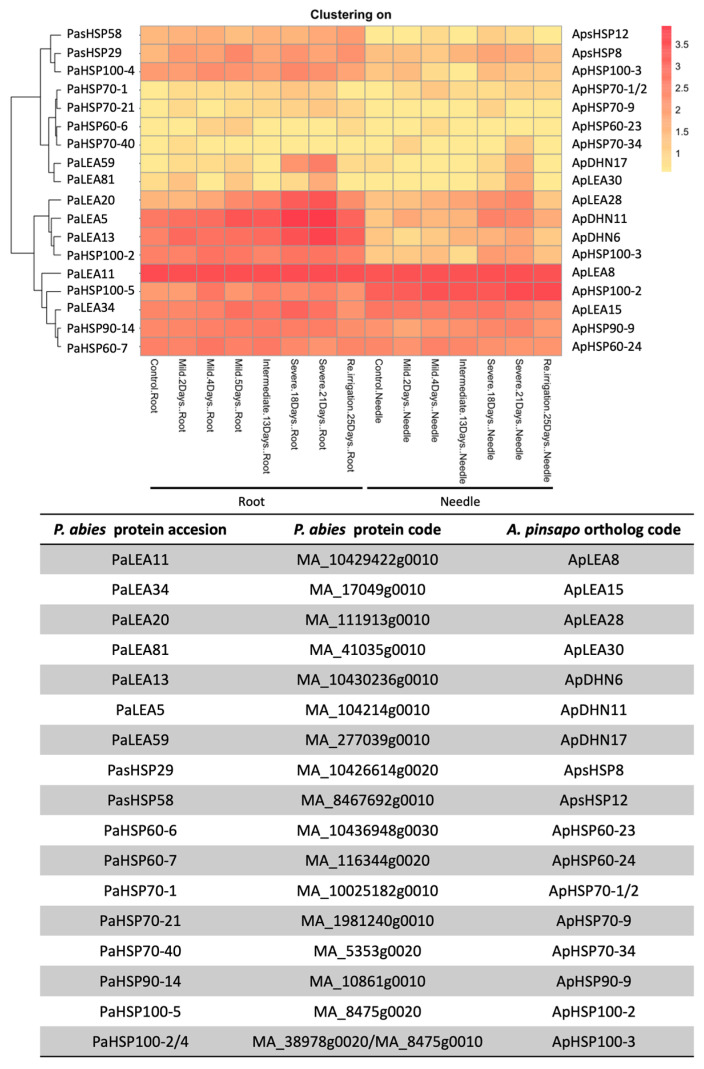
**Selection of *A. pinsapo* candidate genes**. Responses of *P. abies* (Norway spruce) genes to mild, intermediate, and severe drought stress on seedlings, emphasizing reversible physiological changes such as altered stomatal conductance and abscisic acid levels [27]. Upper panel: Heatmap of selected orthologous genes with expression of eight sampling time points (0, 2, 4, 5, 13, 18, 21, and 25 days) in both roots and needles of *P. abies* (left) and *A. pinsapo* (right). Expression values range from 0 to 4, indicated by color gradation from white to red. Lower panel: identification (ID) of candidate genes for both *P. abies* and *A. pinsapo*, including *P. abies* protein accession codes.

**Figure 4 ijms-25-04820-f004:**
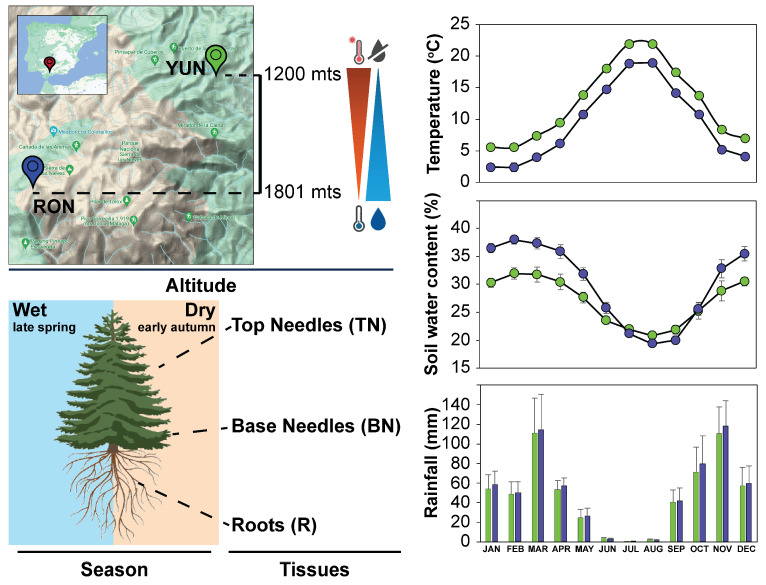
**Localization and climate variables of sampling sites**. On the left, the map localization of the two sampling points at the National Park “Sierra de las Nieves”: Puerto Saucillo (YUN, green) at 1196 m (low water content and high temperature) located in Yunquera municipality, and Cañada del Cuerno (RON, purple) at 1734 m (high water content and low temperature) in the municipality of Ronda. Lower left panel: Different tissues were sampled from pinsapo firs located at different altitudes and contrasting seasonal periods. Collected tissues during the wet (late spring) and dry (early autumn) seasons. Top needles (TNs), base needles (BNs) and roots (R). On the right, seasonal variations in temperature, soil water content, and rainfall at the two sampling sites.

**Figure 5 ijms-25-04820-f005:**
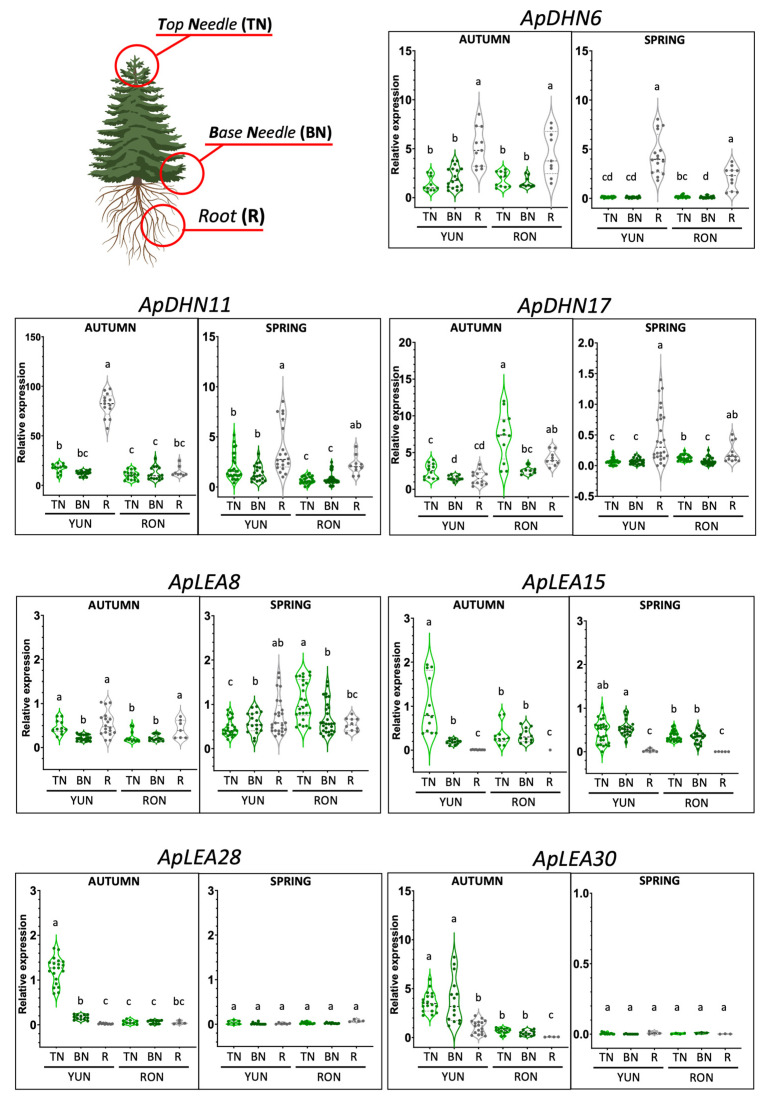
**Expression patterns of pinsapo *LEA* and *dehydrin* genes.** Transcript levels of genes from LEA and dehydrin families in pinsapo firs growing under natural conditions in the forest of “Sierra de las Nieves”. YUN, samples from the location Yunquera; RON, samples from the location Ronda. TN, Top needles; BN, base needles; R, roots. Ten individual trees from each location were analyzed. Each qPCR analysis involved three biological replicates and three technical replicates per sample. Statistical differences were assessed using Past 4.03 ([29]). The initial data analysis included a Krus—all—Wallis test, and in the presence of significant differences, a multiple comparisons test (Dunn posthoc) was applied.

**Figure 6 ijms-25-04820-f006:**
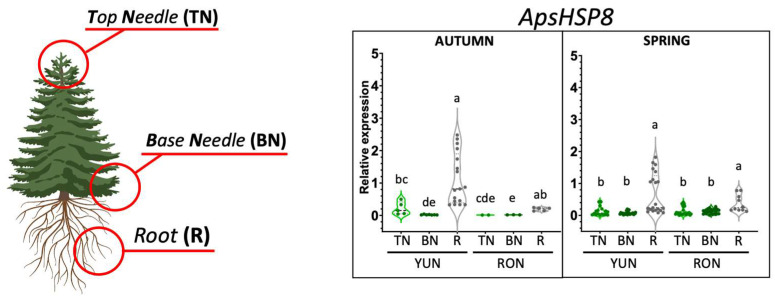
**Expression patterns of pinsapo *HSP* genes.** Transcript levels of genes from HSP families in pinsapo firs growing under natural conditions in the forest of “Sierra de las Nieves”. YUN, samples from the location Yunquera; RON, samples from the location Ronda. TNs, top needles; BNs, base needles; R, roots. Ten individual trees from each location were analyzed. Each qPCR analysis involved three biological replicates and three technical replicates per sample. Statistical differences were assessed using Past 4.03 [29]. The initial data analysis included a Kruskall–Wallis test, and in the presence of significant differences, a multiple comparisons test (Dunn posthoc) was applied.

**Figure 7 ijms-25-04820-f007:**
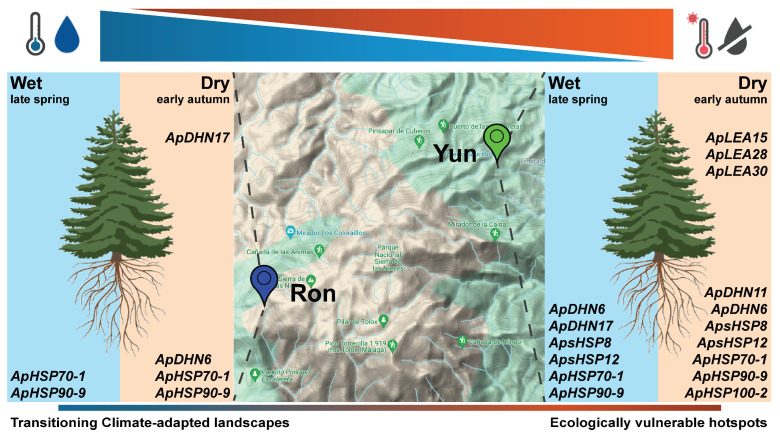
**Variations in stress-related genes in pinsapo forests.** Schematic representation summarizing the expression profiles of *LEA*, *Dehydrins* and *sHSP* genes in pinsapo firs growing at two forest locations under contrasting temperatures and water deficit conditions.

## Data Availability

The original contributions presented in the study are included in the article/Appendix A, further inquiries can be directed to the corresponding author/s.

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
