# Peer review of "Under Stress: Searching for Genes Involved in the Response of Abies pinsapo Boiss to Climate Change"

_ijms, 2024, doi:10.3390/ijms25094820_

Round 1
Reviewer 1 Report
Comments and Suggestions for Authors
Dear Authors,
The article is about genes involved in the response of Abies pinsapo Boiss to climate change, this article analyzes late embryogenesis abundant proteins (LEA) and heat shock proteins (HSPs) gene families of A. pinsapo. Overall, the manuscript is easy to follow. Graphically, the article is clear and adequately prepared. The conducted research methodologically meets the assumed goal. However, some details of the related contents need further clarify, specific comments were listed below:
1. Line 18: "Abies pinsapo" here should be abbreviated.
2. A figure caption should be added to all your figures.
3. L464: The root was collected in the method, and the main part of the root should be described in detail.
4. After RNA extraction, gel electrophoresis should be performed to determine that RNA has not degraded and can be used for subsequent experiments.
5. Some references cited in the manuscript have spaces, while others do not, such as lines 480-488, please ensure consistency.
6. L512-514: What are the parameters used to build the phylogenetic tree?
7. Please note that all scientific names in the references should be italicized.
8. The references should be consistent and meet the requirements of the journal.
9. L85-87 should be reformulated and clarified.
Author Response
Dear Reviewer 1,
Thank you sincerely for your kind review of our article. We have carefully noted your constructive feedback regarding certain details of the related content that may require further clarification. Your specific comments have been duly considered, and we are committed to addressing each point comprehensively in our revision.
- Line 18: "Abies pinsapo" here should be abbreviated.
We have corrected it accordingly.
- A figure caption should be added to all your figures.
Figure captions have been added to all figures.
- L464: The root was collected in the method, and the main part of the root should be described in detail.
Thanks for noticing it. The method for collecting the root has been elaborated upon, specifying the main part harvested (see line 472).
- After RNA extraction, gel electrophoresis should be performed to determine that RNA has not degraded and can be used for subsequent experiments.
A detailed description of gel electrophoresis has been conducted to ensure RNA integrity, with detailed descriptions provided in lines 481-483.
- Some references cited in the manuscript have spaces, while others do not, such as lines 480-488, please ensure consistency.
Inconsistencies in referencing style regarding spaces have been rectified. Thanks for noticing it.
- L512-514: What are the parameters used to build the phylogenetic tree?
Parameters used for building the phylogenetic tree have been included in the Material and Methods section under "Phylogenetic Analysis" (lines 525-530).
- Please note that all scientific names in the references should be italicized.
Thanks again for noticing. Scientific names in both references and the main text have been italicized as per your suggestion.
- The references should be consistent and meet the requirements of the journal.
We utilized a reference reference management tool (Zotero) to ensure consistency and adherence to journal requirements.
- L85-87 should be reformulated and clarified.
As kindly requested, the ideas in lines 85-87 have been reformulated and clarified for better understanding, as reflected in lines 90-92.
We believe these revisions strengthen the clarity and rigor of our manuscript. We are grateful for your guidance and remain committed to enhancing the quality of our work.
Reviewer 2 Report
Comments and Suggestions for Authors
Dear Editor,
Please receive my review for the manuscript entitled “Under stress: Searching for genes involved in the response of Abies pinsapo Boiss to climate change by Blanca-Reyes et al., 2024.
The authors investigated the molecular response to climate-change associated stresses in the Spanish fir (A. pinsapo) to identify the molecular mechanisms governing the response to increased temperature and water stress and candidate genes and their differential seasonal expression patterns using needles and roots in two sites.
The manuscript is well-written and comprehensive. In spite of this, the manuscript needs revision in the below items. The authors are invited to address the below items before acceptance. Please let me know if you have questions.
-Abstract:
Provide specific some findings for gene families, like late embryogenesis abundant proteins (LEA) and heat shock proteins 20 (HSPs) that distinct expression patterns between wet and dry seasons. How can we benefit from your findings.
-Results/Materials and Methods:
Since the experiment was conducted in two sites, ANOVA table should show the effect of temperature, drought, location, and their interactions. This will give us an idea about the level of significance of main effect and their interactions with others.
-Discussion:
The authors should address the limitation of the finding in terms that the experiment was conducted one time in each site and not repeated. Replicated the experiment across seasons and location will increase the reliability of the expression response. This should be addressed in the discussion section.
Author Response
Dear Reviewer 2,
Thank you sincerely for your insightful review of our manuscript. We greatly appreciate the time and effort you've invested in providing constructive feedback. Your detailed comments have been invaluable in guiding our revisions to improve the clarity and robustness of our study.
We have carefully considered each of your suggestions and have made the following revisions:
-Abstract:
Provide specific some findings for gene families, like late embryogenesis abundant proteins (LEA) and heat shock proteins 20 (HSPs) that distinct expression patterns between wet and dry seasons. How can we benefit from your findings.
Thank you for the suggestion. We have clarified and reformulated our abstract, adding further insights from our results, as well as the implications of our findings for studying environmental conditions related to Abies pinsapo by elucidating the molecular mechanisms underlying their response to climate-driven stresses.
-Results/Materials and Methods:
Since the experiment was conducted in two sites, ANOVA table should show the effect of temperature, drought, location, and their interactions. This will give us an idea about the level of significance of main effect and their interactions with others.
We have successfully conducted an ANOVA analysis incorporating the effects of temperature, drought, location, and their interactions, as suggested. This comprehensive approach allowed us to assess the significance of the main effects of temperature and drought, as well as their interactions with location. By examining the ANOVA table, we were able to gain valuable insights into the relative importance of each factor and their combined effects on the experimental outcomes. Thanks for the suggestion.
-Discussion:
The authors should address the limitation of the finding in terms that the experiment was conducted one time in each site and not repeated. Replicated the experiment across seasons and location will increase the reliability of the expression response. This should be addressed in the discussion section.
We have duly considered the reviewer's suggestion and have addressed it in the discussion section of our manuscript, specifically in lines 409-412. In this section, we emphasize the importance of increasing the reliability of our expression response by conducting replicated experiments across seasons and locations. Moving forward, we strongly advocate for the inclusion of replicated experiments in future studies to further validate and provide a more comprehensive understanding of the gene stress-related responses to strengthen our current findings.